# The Use of Response Surface Methodology as a Statistical Tool for the Optimisation of Waste and Pure Canola Oil Biodegradation by Antarctic Soil Bacteria

**DOI:** 10.3390/life11050456

**Published:** 2021-05-20

**Authors:** Khadijah Nabilah Mohd Zahri, Azham Zulkharnain, Claudio Gomez-Fuentes, Suriana Sabri, Khalilah Abdul Khalil, Peter Convey, Siti Aqlima Ahmad

**Affiliations:** 1Department of Biochemistry, Faculty of Biotechnology and Biomolecular Sciences, Universiti Putra Malaysia (UPM), Serdang 43400, Selangor, Malaysia; khadijahnabilah95@gmail.com; 2Department of Bioscience and Engineering, Shibaura Institute of Technology, College of Systems Engineering and Science, 307 Fukasaku, Minuma-ku, Saitama 337-8570, Japan; azham@shibaura-it.ac.jp; 3Department of Chemical Engineering, Universidad de Magallanes, Avda. Bulnes 01855, Punta Arenas, Chile; claudio.gomez@umag.cl; 4Center for Research and Antarctic Environmental Monitoring (CIMAA), Universidad de Magallanes, Avda. Bulnes 01855, Punta Arenas, Chile; 5Department of Microbiology, Faculty of Biotechnology and Biomolecular Sciences, Universiti Putra Malaysia (UPM), Serdang 43400, Selangor, Malaysia; suriana@upm.edu.my; 6Faculty of Applied Sciences, School of Biology, Universiti Teknologi MARA, Shah Alam 40450, Selangor, Malaysia; khali552@uitm.edu.my; 7British Antarctic Survey, NERC, High Cross, Madingley Road, Cambridge CB3 0ET, UK; pcon@bas.ac.uk; 8National Antarctic Research Centre, B303 Level 3, Block B, IPS Building, Universiti Malaya, Kuala Lumpur 50603, Malaysia

**Keywords:** waste canola oil, pure canola oil, Antarctic bacterial consortium, one-factor-at-a-time, response surface methodology

## Abstract

Hydrocarbons can cause pollution to Antarctic terrestrial and aquatic ecosystems, both through accidental release and the discharge of waste cooking oil in grey water. Such pollutants can persist for long periods in cold environments. The native microbial community may play a role in their biodegradation. In this study, using mixed native Antarctic bacterial communities, several environmental factors influencing biodegradation of waste canola oil (WCO) and pure canola oil (PCO) were optimised using established one-factor-at-a-time (OFAT) and response surface methodology (RSM) approaches. The factors include salinity, pH, type of nitrogen and concentration, temperature, yeast extract and initial substrate concentration in OFAT and only the significant factors proceeded for the statistical optimisation through RSM. High concentration of substrate targeted for degradation activity through RSM compared to OFAT method. As for the result, all factors were significant in PBD, while only 4 factors were significant in biodegradation of PCO (pH, nitrogen concentration, yeast extract and initial substrate concentration). Using OFAT, the most effective microbial community examined was able to degrade 94.42% and 86.83% (from an initial concentration of 0.5% (*v*/*v*)) of WCO and PCO, respectively, within 7 days. Using RSM, 94.99% and 79.77% degradation of WCO and PCO was achieved in 6 days. The significant interaction for the RSM in biodegradation activity between temperature and WCO concentration in WCO media were exhibited. Meanwhile, in biodegradation of PCO the significant factors were between (1) pH and PCO concentration, (2) nitrogen concentration and yeast extract, (3) nitrogen concentration and PCO concentration. The models for the RSM were validated for both WCO and PCO media and it showed no significant difference between experimental and predicted values. The efficiency of canola oil biodegradation achieved in this study provides support for the development of practical strategies for efficient bioremediation in the Antarctic environment.

## 1. Introduction

The use of edible oils, in particular different types of vegetable oil, has increased by at least a million metric tonnes per year worldwide over the last 10 years [1]. According to statistics on the consumption of vegetable oil, 80% is used in cooking as frying oil, salad oils, shortening, spreads, and in steaming and boiling [2]. It is generally known that cooking oil can be utilised as a heat transfer medium and contributes flavour and texture to foods due to its chemical properties. Unfortunately, the disposal of waste cooking oil into the sewage system and the natural environment is routine and could cause an unpleasant odour in the surroundings, where oxidative degradation of canola oil (waste) produces several volatile products that cause the soil to give off an unpleasant odour. Threats to the environment are exacerbated by the presence of toxic organic compounds, such as acrylamide, aldehyde and 4-hydroxymethylfurfural which are formed in heated cooking oil and are known to have mutagenic and carcinogenic effects [3,4]. Vegetable oil can also lead to a devastating physical effect on organisms in the environment and destroy future and existing food supplies, along with breeding animals and their habitats.

The Antarctic is known as a perfect place for judging the spread of global pollutants and a sensitive indicator of dramatic global change [5]. This shows that Antarctic research is important to the environmental, scientific and intrinsic values worthy of protection for the future of the world [6]. The common dietary fat widely used in most Antarctic research stations reported in past research is canola oil, which is able to maintain its physical properties in cold conditions and is usually used by the four season’s countries [2,7]. Based on regulations detailed in the Protocol on Environmental Protection to the Antarctic Treaty, waste cooking oil should be stored in secure containers before being shipped out of the Antarctic Treaty area for disposal, in order to minimise the risk of dispersal into the environment [8]. However, it is virtually impossible to avoid some waste oil entering the environment with the permitted release of ‘grey water’ from stations and ships, while the risk of spillage events remains during storage and transport of oils.

The use of microbial groups in the degradation of pollutants is known as bioremediation. Microorganisms play a critical role in bioremediation due to their abundance, diversity and community structure in contaminated environments. Microbial degradation of vegetable oils provides a viable alternative compared to conventional means of disposal [9] and can be more efficient and cost effective compared to chemical or physical techniques [10]. Extracellular bacterial enzymes allow the degradation of complex molecules to less or non-toxic forms that are safe to be discharged into the environment [11,12]. The biodegradation of vegetable oil from triglycerides produces free fatty acids and acetate which can act as major sources of nutrients allowing microbes to proliferate [13], and in which the short carbon chain or simple compound is used by the bacteria in their growth and metabolism. The use of combinations or consortia of different bacteria has been shown to be more effective in the degradation of pollutants compared to a single species or strain [14,15], which may be due to the synergistic effect among the members of specific consortia [16], since the survival and stability of most bacteria are more successful within a community [17,18].

The efficiency of bioremediation processes depends on multiple biological and environmental factors, including temperature, availability of nutrients and pH [19]. Understanding the influence of variation in these factors is important to provide the optimal conditions that are essential for microbial metabolism in order to degrade the targeted pollutant in the environment [10]. A number of tropical bacteria have been identified with potential in the biodegradation of waste cooking oils, grease and kitchen waste [20]. Nzila et al. [17] reported that a bacterial consortium (including *Stenotrophomonas rhizophila*, *Sphingobacterium* sp., *Pseudomonas libanensis*, *P. poae* and *P. aeruginosa*) isolated from wastewater in a Saudi Arabian city could degrade 90% of oil within 7–9 days. Fewer studies have addressed the biodegradation of vegetable oil by bacteria native to cold environments [21], currently limited to the single strain *Rhodococcus* sp. AQ5-07 isolated from Antarctic soil [22,23], and no studies have addressed biodegradation by bacterial consortia in such environments. Cold tolerant microorganisms are of potentially great importance in the development of bioremediation approaches for degrading hydrocarbons from contaminated soil in polar regions, as they can sustain metabolic activity under extreme conditions [24,25]. Such bacteria can be either psychrophilic or psycho-tolerant. A number of cold-adapted Antarctic bacteria capable of degradation of hydrocarbons have been reported, including members of the genera *Arthrobacter, Rhodococcus, Pseudomonas, Acinetobacter, Bacillus* and *Sphingomonas* [26,27,28,29,30].

This study aims to optimise the growth and conditions of Antarctic bacterial consortia for WCO and PCO degradation. The use of the classical and statistical method was applied in this study, which can provide positive results with better degradation, besides specifying how the community composition can best be complemented to enable the mineralisation of pollutants [31]. Considering that there is no study yet reported on the pollution of vegetable oil in the Antarctic, the potential application for further study using the findings from this project is in the bioremediation of waste cooking oil from the storage containers using a bioreactor tank in the Antarctic itself to reduce the possibilities of an oil spill during transportation, and to lower the cost of transportation of the ship from the Antarctic. In the meantime, the lipid-rich wastewater treatment, specifically kitchen wastewater, could also be applied in the Antarctic using the optimum conditions acquired in this study, as the condition is easily monitored and controlled during the bioremediation processes.

## 2. Materials and Methods

### 2.1. Sample Collection and Preparation

Twenty-eight soil samples were obtained in February 2019, using sterile spatulas (approximately 15 g of soil each) around the Chilean General Bernardo O’Higgins Riquelme Station on the Trinity Peninsula, north-west Antarctic Peninsula (63°19′15″ S 57°53′55″ W). Soil sampling locations were separated by at least 10 m and the depth of the soil sample collected was around 2 to 5 cm. Then, the samples were placed in sterile 50 mL Falcon tubes and stored at 4 °C for 2 months after collection and during transport to Malaysia. A sub-sample (c. 0.5 g) from each soil sample was inoculated separately into nutrient broth, NB (Friendmann Schmidt, Germany) and incubated at 10 °C on an orbital shaker at 150 rpm for 3–4 days before being sub-cultured into fresh NB three times. WCO and PCO (Belmont, Chile) were obtained directly from the station’s kitchen (no information was available about the frequency of use of the WCO). The WCO was sterilised by filter sterilisation. The ability to degrade both PCO and WCO efficiently is important because the chemical composition of the oils might be different due to repeated exposure to heat during the cooking process.

### 2.2. Screening of WCO and PCO Degradation by Bacterial Consortia

The ability of the bacterial consortia obtained from the different soil samples to degrade WCO and PCO was screened by growing 1 mL of standardised resting cells in 50 mL modified minimal salt (MSM) medium (0.5 g L^−1^ MgSO_4_.7H_2_O, 0.56 g L^−1^ KH_2_PO_4_, 4.74 g L^−1^ K_2_HPO_4_, 1 g L^−1^ (NH_4_)_2_SO_4_ and 2.5 g L^−1^ NaCl) supplemented with 1% initial concentration of oil [22]. Resting cells were obtained by growing the bacterial cultures in NB at 10 °C and 150 rpm for 4 d. The cells were then harvested by centrifugation at 7000× *g* at 4 °C for 10 min. The cell pellet was washed twice with phosphate buffer saline (PBS) at pH 7.4. The resting cells acquired, and the inoculum size of the bacterial samples, were standardised to an optical density of 1.0 ± 0.01 at a wavelength of 600 nm.

Biodegradation of WCO and PCO was determined using the gravimetric method [32]. Lipids were extracted by transferring the culture media into separating funnels with the addition of n-hexane at a 1:1 ratio of medium to solvent. The funnels were vigorously shaken, after which the organic phase was allowed to separate for 15 min [33]. The top organic layer was then transferred into a glass dish and concentrated under a fume hood. The amount of oil remaining was determined gravimetrically, and the degradation rate was expressed as the biodegradation efficiency (BE) relative to the abiotic loss of canola oil in experimental controls, using Equation (1) [16]:(1)$$\mathrm{BE}\left(\%\right)=(\frac{\begin{array}{c} \mathrm{Weight}\;\mathrm{of}\;\mathrm{residual}\;\mathrm{canola}\;\mathrm{oil}\;(\mathrm{abiotic}\;\mathrm{control})-\;\\ \mathrm{Weight}\;\mathrm{of}\;\mathrm{residual}\;\mathrm{canola}\;\mathrm{oil}\;(\mathrm{sample}) \end{array}}{\mathrm{Original}\;\mathrm{weight}\;\mathrm{of}\;\mathrm{canola}\;\mathrm{oil}\;\mathrm{introduce}})\;\times 100$$

### 2.3. Optimisation of Bacterial Growth and Canola Oil Degradation through One-Factor-at-a-Time (OFAT)

The sample consortium showing highest degradation (coded as BS14) was cultivated by adding 1 mL of bacterial sample to 50 mL of MSM (see Section 2.2) under different conditions, namely salt concentration, pH, nitrogen source, temperature, yeast extract concentration and initial oil concentration. The original MSM values and conditions were used as the constant level for non-optimised factors, maintaining the parameters that showed the highest canola oil degradation for the factors tested subsequently. Bacterial growth and oil degradation were assessed by measuring optical density at 600 nm and gravimetry, respectively [34]. Salt concentration was varied from 0 to 1.5% (*w*/*v*). The samples were incubated at 10 °C on an orbital shaker at 150 rpm for 7 d, and bacterial growth was determined every 24 h. The influence of pH was tested across the range pH 5.5 to 8.5 using 50 mM acetate buffer (pH 5.5 and 6.0), phosphate buffer (pH 6.0, 6.5, 7.0 and 7.5) or Tris-HCl (pH 7.0, 7.5, 8.0 and 8.5). The influence of nitrogen source was tested using KNO_3_, NH_4_Cl, NH_4_NO_3_, NaNO_3_, (NH_4_)_2_SO_4_, (NH_4_)_2_HPO_4_ and (NH_4_)_6_Mo_7_O_24_ [35]. Different N sources were tested in this study since not all can serve equally well as nutrients for different bacteria [36]. The concentration of the nitrogen source used is an important factor in the design of bacterial bioremediation approaches [37]. Then, using the N source giving the highest percentage of degradation, the influence of source concentration was tested across the range 0 to 3 g/L. The influence of temperature tested at 5, 10, 15, 20 and 25 °C, and the influence of yeast extract concentration, was tested at 0, 0.05, 0.25, 0.50, 0.75, 1.00 and 1.25 g/L. Trials of yeast extract were carried out because it contains proteins, short chain peptides, carbohydrates, free amino acids and nucleic acids, the combination of which could influence the efficiency of hydrocarbon degradation. Finally, the influence of initial concentration of WCO or PCO was tested between 0% and 7% (*w*/*v*).

### 2.4. Statistical Optimisation using Response Surface Methodology (RSM)

A wide range of parameters and their interaction effects can play a role in the optimisation process using methodologies such as RSM [38]. To help simplify this, the initial use of a classical method such as OFAT can help in narrowing down the optimum range for each factor tested before proceeding to RSM. RSM was carried out based on surface placement with the key objectives being to study the topography of the response surface including the local maximum, local minimum and ridgelines and to identify the region (combination of conditions) where the optimum response occurs [39]. The Plackett-Burman design (PBD) was used to screen and remove factors with no significant influence on the degradation of WCO or PCO before optimising the response using central composite design (CCD). Statistical optimisation was carried out using Design Expert version 6.0.8 software (Stat-Ease Inc., Minneapolis, MI, USA).

#### 2.4.1. Screening for Significant Factors Using Plackett-Burman Design (PBD)

All elements from OFAT were tested in the PBD at high (+1) and low (−1) levels selected from the highest response observed under the OFAT approach. All significant factors for biodegradation of WCO and PCO were evaluated and used for further optimisation using CCD. The maximum and minimum levels of the factors that affected WCO and PCO degradation are presented in Table 1.

Statistical analysis was performed using 12 experimental runs for each of WCO and PCO. A first order polynomial with linear process order was selected incorporating all individual significant factors affecting canola oil degradation using Equation (2) [40]:(2)$$E\;\mathrm{xi}=\left(\frac{\sum \mathrm{Mi}+-\mathrm{Mi}-}{N}\right)$$
where *E* xi is the effect of the tested factors, Mi+ and Mi− are the response (canola degradation) of trials, and *N* is the total number of trials divided by 2. The positive and negative signs indicate the high and low experimental values, respectively, for each of the factors (Table 1).

#### 2.4.2. Optimisation of WCO and PCO Degradation Using Central Composite Design (CCD)

CCD, categorised as a set of advanced ‘design of experiments’ (DoE) methods, allows the evaluation of the effects of multiple factors and their interactions. CCD allows optimisation of pairs of factors [41]. Three levels were used for each factor, and the response was estimated using a quadratic model for the type of process order (Equation (3)) [42]:(3)$$Y=\beta_{0}+\overset{k}{\underset{i=1}{\sum }}\beta_{i}x_{i}+{\left(\overset{k}{\underset{i=1}{\sum }}\;\beta_{ii}x_{i}\right)}^{2}+\overset{n-1}{\underset{i=1}{\sum }}\overset{k}{\underset{j=i+1}{\sum }}\beta_{ij}x_{i}x_{j}$$
where Y is the response (canola oil degradation), *β_i_* is a linear term, *β_ii_* is a quadratic term and *β_ij_* is an interaction term.

Significant factors generated from PBD were used in CCD for both WCO and PCO degradation. The low and high levels of experimental values were used, as given in Table 1. Three-dimensional response surface plots were constructed to determine the interaction among the factors considering the statistical parameters obtained, including the R^2^ and the lack of fit of the model [43].

### 2.5. Statistical Analysis

All experiments were carried out in triplicate, with experimental error indicated by the standard error of the mean (SEM). Data were analysed using one-way ANOVA followed, where significant, by post hoc pairwise Tukey’s tests (GraphPad InStat, San Diego, CA, USA), with *p* < 0.05 being accepted as statistically significant. ANOVA for the experimental design in RSM was carried out using the RSM software (see Section 2.3) with *p* < 0.05 accepted as statistically significant.

## 3. Results

### 3.1. Screening of WCO and PCO-Degrading Antarctic Bacterial Consortia

The bacterial consortia obtained by culturing from all 28 soil samples were tested for their ability to degrade WCO and PCO over a 5 days period. Most of the consortia were able to degrade >10% of the initial amount of WCO or PCO. The highest degradation of WCO was achieved by the consortium from sample BS23 (44.83%), followed by BS12, BS11 and BS14 (Figure 1a). The highest degradation of PCO was achieved by the consortium from sample BS16 (51.24%), followed by BS14 and BS3 (Figure 1b). Although BS23 achieved the greatest degradation of WCO, its ability to degrade PCO was lower than BS14. The consortium from sample BS14 (isolated from location 62°46′31″ S 81°18′48″ E), achieving degradation of 36.73% and 50.59% of WCO and PCO, respectively, was therefore selected for further optimisation using OFAT and RSM. Each consortium tested is likely to have different bacterial diversity and therefore different degradation capabilities. The consortium obtained from sample BS14 performed well compared to those from most other samples, clearly being able to produce the main enzymes involved in the process of breaking down oil and lipid [44,45].

The consortium from sample BS14 showed both high bacterial growth and a high degradation rate. Generally, the rate of degradation of contaminants is high when the bacterial growth of the samples increases [46,47]; nevertheless, some of the consortia examined here achieved oil degradation even with low bacterial growth rates (Figure 1).

Normally, the group of identified Antarctic bacteria in biodegraded hydrocarbon comes from the same group of bacteria. Since cooking oil is classified as a hydrocarbon, the probability of Antarctic bacteria species occurring in this study is likely to be similar to the bacteria that have been identified as revealed in the past few decades. *Pseudomonas* and *Rhodococcus* genus are the best known bacteria with the capability of breaking down hydrocarbon; for instance, *Pseudomonas* sp. strain ST41 isolated from Signy Island, *Pseudomonas* sp. strain Ant 9 (Ross Island), *Pseudomonas* sp. strain LCY 16 (King George Island), *Rhodococcus* sp. strain AQ5-07 (King George Island), *Rhodococcus* sp. strain DM1-21 (Marambio Island), *Rhodococcus* sp. strain JG-3 (Ross Island) and *Rhodococcus* sp. (South Shetland Island) (South Shetland Island) [22,27,29,48,49,50,51,52]. These Actinobacteria have been widely reported in most metagenomics studies, specifically in Antarctic Peninsula areas, including in the Korean Antarctic Research Station and Alexander Island [53,54,55]. There are also other bacteria (less common) that have been listed as hydrocarbon-degrading bacterial species isolated from Antarctica and summarised by Wong et al. (2021): *Arthrobacter* spp. strain AQ5-05 (King George Island), *Sphingomonas* sp. strain Ant 17, *Sphingobium xenophagum* strain D43FB (King George Island), *Planococcus* sp. strain NJ41 and *Shewanella* sp. strain NJ49 (Antarctic Ocean) [56].

The degradation activity was observed to be different between WCO and PCO media for the same bacteria consortium, which might be caused by the different compositions of the substrates. The common fatty acid composition of canola oil (PCO) comprises 6% saturated fatty acid (palmitic and stearic) and 92% unsaturated fatty acid (oleic, linoleic and linolenic) [2]. On the other hand, an increasing number of saturated fatty acids in waste cooking oil compared to the unused oil has been reported by Knothe and Steidley [57], where the composition of steric and oleic compound increases along with the decreasing number of linoleic and linolenic fatty acids in waste oil. Lots of volatile organic compounds have also been found in the waste oil, which include acetaldehyde, methylamine, N,N,-dimethyl-octane, pyrazine, formamide, N,N-dimethylacetamide and furfural [58]. These compounds might have been produced by chemical changes that occurred in waste cooking oil during the thermal heating processes (cooking), making the PCO more complex compared to the WCO.

### 3.2. Optimisation of Consortium BS14 Growth and Canola Oil Degradation Using One-Factor-at-a-Time

Overall, the optimum conditions identified using OFAT for WCO degradation were 0% (*w*/*v*) NaCl, pH 7.5, 1 g/L ammonium sulphate, 10 °C, 1.25 g/L yeast extract and 0.5–1% (*v*/*v*) initial substrate concentration. For PCO degradation, only the optimum nitrogen source concentration differed from that for WCO degradation, at 0.5 g/L (NH_4_)_2_SO_4_.

Sodium chloride (NaCl) was used to determine the influence of salt concentration on bacterial growth and degradation of WCO and PCO (Figure 2a and (Figure 3a). Both bacterial growth and oil degradation rates rapidly decreased as the concentration of NaCl increased above 0.25% (*w*/*v*). Degradation of both WCO and PCO was greatest at 0% NaCl, achieving 44.88% and 78.66%, respectively. Higher bacterial growth was also observed at 0% and 0.25% salt concentrations. These data are consistent with previous studies on the biodegradation of contaminants by Antarctic soil bacteria, which also documented greater growth and degradation at low salt concentrations. For instance, Zakaria et al. (2019) showed that an Antarctic isolate of *Rhodococcus boikonurensis* was able to degrade an initial concentration of 0.2 g/L of phenol with up to 100% degradation within 48 h at a salt concentration of 0.01% (*w*/*v*) of NaCl [59]. Similarly, *Arthrobacter* sp. strain AQ5-05 showed the greatest hydrocarbon degradation rate at low salt concentration (0 to 3% *w*/*v*) through statistical optimisation [60]. Salt concentrations in soils of the sampling area, despite its coastal location, can be categorised as low (0–0.3%) [61]. Impacts of high salt concentrations include osmotic effects and reduced enzyme activity, soil microbial biomass and bacterial growth rate [62,63].

Bacteria are generally sensitive to pH variation in their environment [64]. The data obtained here (Figure 2b and (Figure 3b) showed bacterial consortium growth and the degradation of both WCO and PCO were optimum at close to neutral pH and strongly inhibited at lower and higher pH. Both growth rate and WCO and PCO were most strongly inhibited by acidic conditions. At the optimum pH the degradation achieved was 46.06% for WCO and 81.67% for PCO. Ibrahim et al. (2020) reported that the Antarctic *Rhodococcus* sp. strain AQ5-07 was most effective at degrading WCO at pH 7.5 [22]. The same bacterial strain could also effectively degrade diesel at pH 7 [29].

Nitrogen is a major element supporting function in microbial cells, being a key constituent of amino acids, nucleotides and all proteins [65]. Inorganic nitrogen sources were trialled in this study, since the carbon source was solely provided from the WCO or PCO substrate. Use of (NH_4_)_2_SO_4_ led to high degradation of both WCO and PCO, while use of NH_4_Cl, NH_4_NO_3_ and (NH_4_)_2_HPO_4_ also led to high degradation (>50%) of PCO (Figure 2c and Figure 3c). Bacterial growth was also high when supplied with (NH_4_)_2_SO_4_. NH_4_Cl stimulated the greatest bacterial growth, but had less influence on degradation rate compared to (NH_4_)_2_SO_4_. It is possible that the different nitrogen sources may have stronger influences on cellular products and pathways other than lipolytic enzymes [66]. Studies of Antarctic bacteria have previously reported that (NH_4_)_2_SO_4_ is the most effective nitrogen source. For example, the cold-adapted Antarctic soil bacteria, *Arthrobacter* sp. strain AQ5-05, *Arthrobacter* sp. strain AQ5-06 and *Rhodococcus* sp. strain AQ5-07, exhibited maximum degradation of phenol when provided with ammonium sulphate [67]. *Arthrobacter* sp. strain AQ5-15 also achieved greater degradation of phenol when using ammonium sulphate as nitrogen source [68]. Ammonium sulphate has the practical advantages of being low-cost, readily available and suitable for application on a large scale compared to other nitrogen sources [69,70] Ammonium bisulphate or ammonium sulphate have also been suggested to be the dominant forms present in Antarctic atmospheric studies, especially in coastal regions with high concentrations of marine vertebrates [71,72,73].

Bacterial growth and oil degradation rates were high at concentrations of (NH_4_)_2_SO_4_ of up to 1 g/L for WCO and 0.5 g/L for PCO (Figure 2d and Figure 3d). Maximum degradation was achieved at the lowest concentration trialled, decreasing by around 10% as the concentration was progressively increased. Previous studies have similarly reported that only low concentrations of ammonium sulphate were required to support bacterial growth and substrate degradation. *Arthrobacter* sp. strains AQ5-05 AQ5-06 were able to degrade more than 40% of diesel when provided with 0.4 g/L ammonium sulphate [30]. As shown in Figure 2d (WCO medium), the BS14 bacterial consortium required a higher concentration of nitrogen source for optimum growth and degradation compared to that required in PCO medium (Figure 3d). This might be due to the production of by-products from thermal degradation of the oil (from heat during cooking), including volatile and toxic compounds [4]. Swiecilo and Zych-Wezyk (2013) stated that unfavourable environmental conditions lead to the activation of stress response mechanisms [74]. Large energy expenditure is required for bacteria to activate the adaptive mechanisms to synthesise defence molecules. This may explain why the BS14 consortium in WCO medium required a greater N concentration.

Oil degradation and bacterial growth were tested at temperatures between 5 °C and 25 °C. Generally, degradation of WCO and PCO declined as temperature increased (Figure 2e and Figure 3e). Degradation of WCO was significantly greater at 10 °C than 5 °C and 15 °C, but there was no significant difference in the degradation of PCO between 10 °C and 15 °C. The temperature obtained during soil sample collection ranged from −0.1 °C to 8.5 °C using a thermocouple thermometer. However, the result showed that the bacteria consortia preferred the high temperature to the actual conditions in the Antarctic, which was at 10 °C. Numerous studies have also reported optimum activity of cold-adapted bacteria around 10–15 °C [22,30,68,75,76]. While the degradation rate was lowest at 25 °C, bacterial growth was high (Figure 2e and Figure 3e). This may indicate that the subset of bacteria capable of surviving at this high temperature did not have the ability to degrade canola oil. Although the optimum temperature in degradation of WCO and PCO is relatively high from the actual value, the conditions of the Antarctic today also have an impact future application. According to the World Meteorological Organisation (2020), the new record temperature in the Northern Antarctic Peninsular was 18.4 °C during February 2020 due to global warming effects [77]. Therefore, the highest activity obtained at 10 °C is acceptable for the application of this study in the Antarctic.

High concentrations of yeast extract increased both bacterial growth and WCO and PCO degradation (Figure 2f and Figure 3f). Yeast extract acts as a primary growth substrate for the bacteria to co-oxidise the oil [78]. Previous studies have shown that the addition of yeast extract promoted the biodegradation of oil hydrocarbons and the growth of bacterial consortia [79]. Yeast extract provides both macro- and micro-nutrients (such as metal ions) as well as vitamins and amino acids which are essential for bacterial growth and metabolism [80,81]. About 0.014 g/L yeast extract led to maximum diesel hydrocarbon degradation by *Acinetobacter beijerinckii* strain ZRS [80], and *Pseudomonas* sp. sp48 required 5 g/L yeast extract to achieve maximum biodegradation of crude oil [82]. The addition of yeast extract led to a more significant effect on the degradation of WCO compared to PCO. As noted above, the chemical changes accumulating in WCO as a result of repeated heating will include various chemical compounds from oxidation, hydrolysis and peroxidation processes, including acrylamide, fatty acids and aldehydes [4,83,84].

High initial concentration of either WCO or PCO inhibited the bacterial consortium’s degradation ability, although bacterial growth was maintained (Figure 2g and Figure 3g). There was no significant difference in degradation at 0.5% and 1% initial concentration of either substrate. Only a single study has reported biodegradation of WCO using a culture of the Antarctic soil bacterium *Rhodococcus* sp. strain AQ5-07 [22]. The methodology differed from the current study, in particular using a much larger bacterial inoculum, which restricts comparison between the two studies. However, that study achieved greater degradation at an initial concentration of 3% WCO over 3 days, compared with around 40% over 7 days in the current study. The inhibition of degradation associated with high concentrations of substrate may indicate suppression of key enzymes [85]. This is known to be the case especially in the biodegradation of hydrocarbons that are composed of aromatic elements [86]. The ability of the bacteria to maintain growth when a high initial oil concentration is provided might result from the presence of yeast extract, providing the nutrient source required to fuel growth (rather than oil degradation).

### 3.3. Optimised Bacterial Growth and Degradation by Response Surface Method (RSM)

#### 3.3.1. Plackett-Burman Design (PBD)

All factors examined using OFAT were analysed using the minimum and maximum experimental values generated in 12 runs for each degradation response for each of WCO and PCO degradation. After incubation for 6 d, the highest and lowest degradation percentages were achieved, these being 80.79% (WCO) and 85.32% (PCO), and 4.24% (WCO) and 1.61% (PCO), respectively. The modelled response was statistically significant.

The significance of the factors tested on canola oil degradation is given in Table 2, with all factors being significant in WCO degradation. This confirmed that the design was significant in predicting the effects of the factors on degrading WCO using the BS14 Antarctic bacterial consortia.

The outcome of analyses for PBD for PCO degradation is shown in Table 3, where four factors; B (pH), C ((NH_4_)_2_SO_4_ concentration), E (yeast extract concentration) and F (initial PCO concentration) had a significant influence on biodegradation of PCO, with A (salinity) and D (temperature) not being significant. The insignificant factors were therefore fixed at the optimum condition identified through OFAT.

#### 3.3.2. Central Composite Design (CCD)

The results of the second-order CCD experimental design were evaluated for optimising WCO and PCO degradation using the BS14 bacterial consortium. Equation (4) includes the coded significant factors and interactions influencing the degradation of WCO, and Equation (5) likewise for degradation of PCO.
(4)$$Y\;\left(\%\right)=57.29+2.40E\;-21.20F\;-2.26E^{2}+2.97\mathrm{DF}$$
(5)$$\begin{array}{cc} & Y\;\left(\%\right)=67.03-4.81A\;-4.25D\;-7.65A^{2\;}-7.28B^{2}-0.6054C^{2\;}\\ & -10.59D^{2}+9.23\mathrm{AD}\;-2.62\mathrm{BC}-5.46\mathrm{BD}\; \end{array}$$
where *Y* is the response (canola oil degradation) and the coded variables in the equations indicate the significant factors identified from PBD (Table 4 and Table 5).

The model developed for WCO degradation was highly significant (Table 4). However, only the individual factors E (temperature) and F (WCO concentration) were significant, as was the interaction between (D) temperature and (F) WCO concentration. Although only this one interaction term of the many possible was significant, post hoc Tukey’s pairwise tests between all combinations of these two variables were confirmed as highly significant (*p* < 0.001), confirming the importance of this interaction term. Of all the 81 runs generated for CCD experimental design in WCO degradation, the 35th run of the design (comprising 0.25% (*w*/*v*) NaCl, pH 7, 1 g/L nitrogen source concentration, incubation at 10 °C and 1.25 g/L yeast extract) yielded the highest percentage degradation over the 6 days incubation, with 94.99% oil degradation at 0.5% initial concentration of WCO.

Table 5 shows that the CCD model for PCO degradation was also highly significant. In this case, as well as A and D, all four quadratic terms and three of the interaction terms (AD, BC and BD) were significant. As in the WCO degradation model, the initial substrate concentration showed a significant interaction with the other factors.

In PCO degradation, 30 runs were generated for the CCD experimental design, and the second run yielded the highest degradation percentage (79.77%) of 0.5% initial concentration of PCO over 6 d, at medium pH 7, with 1 g/L (NH_4_)_2_SO_4_ concentration and 0.75 g/L yeast extract concentration.

The contour plots generated from the software in the form of 3D response surfaces describing the pairwise interactions between factors (Figure 4), are useful for identifying the optimum conditions for canola oil biodegradation for application in bioremediation.

In optimising WCO degradation, the effects of temperature (D) and substrate concentration (F) on the response at fixed salinity (A) of 0.13% (*w*/*v*), pH (B) of 7.25, (NH_4_)_2_SO_4_ concentration (C) of 0.75 g/L and yeast extract concentration of 1.00 g/L are shown in Figure 4a. High initial concentration of substrate lowered the degradation of WCO, most likely by inhibiting the enzymes involved in hydrolysing the complex substrate molecules. The form of the interaction between these two factors enabled the bacterial community to degrade 85.88% of oil across a wide range of temperature.

Degradation of PCO showed different patterns of interaction based on the outcomes of CCD. The factors (NH_4_)_2_SO_4_ concentration (B) and yeast extract concentration (C) were fixed at 0.75 g/L and 0.88 g/L, respectively. The response (*Y*) increased as the pH (A) of the medium decreased from 7.25 to 7.00 and rapidly declined as pH increased above 7.25 (Figure 4b). The response decreased as substrate concentration (D) increased, again likely to be due to the inhibition effect of hydrocarbons as a substrate [87]. The interaction between the factors had a significant effect on the response as indicated by the plot and low *p* value (Table 5).

Figure 4c shows the (NH_4_)_2_SO_4_ concentration (B) and yeast extract (C) effects on the PCO degradation response (*Y*) as the nitrogen concentration increased from 0.63 to 0.75 g/L and with increase in temperature. pH (A) was fixed at 7.25 and initial substrate concentration (D) at 1.25% (*v*/*v*), under which conditions the higher concentration of yeast extract increased the degradation rate of PCO by 69.35%. The response (*Y*) increased as the (NH_4_)_2_SO_4_ concentration (B) and initial substrate concentration (D) increased from 0.50 to 0.75 g/L and 0.88 to 1.25% (*v*/*v*), respectively, thereafter decreasing with further increase in concentrations (Figure 4d). A maximum of 68.36% PCO degradation was predicted by the model. Validation of the statistical model was achieved by performing a degradation experiment using the levels of each factor identified in the CCD [88] (Table 6). The experimental and predicted values closely agreed, validating the model.

## 4. Conclusions

An Antarctic soil bacterial consortium had the ability to efficiently degrade both WCO and PCO based on optimisation using both OFAT and RSM methodologies. Significant factors influencing WCO and PCO degradation were determined through PBD. Optimum conditions as predicted from CCD response analysis for WCO biodegradation were 0.13% (*w*/*v*) salinity, pH 7.30, 1.00 g/L (NH_4_)_2_SO_4_ concentration, temperature 13 °C, 1.13 g/L yeast extract concentration and initial WCO concentration 0.5% (*v*/*v*). Optimum conditions for PCO biodegradation were predicted to be pH 7.25, 0.75 g/L (NH_4_)_2_SO_4_ concentration, 1.00 g/L yeast extract and initial PCO concentration 1.25% (*v*/*v*). The optimum conditions applied led to slightly higher degradation of canola oil in OFAT than in RSM, with values for WCO and PCO of 94.42% and 86.63% through OFAT and 94.99% and 79.77% through RSM, respectively. However, the use of RSM had the advantage of reducing the incubation time by one day. Optimisation by RSM also gives better understanding of the interactions among the significant factors and parameters. The ability of such bacterial consortia to degrade WCO and PCO in cold environments may be usefully applied in bioremediation approaches as the optimum temperature for this consortium was between 10 to 15 °C.

## Figures and Tables

**Figure 1 life-11-00456-f001:**
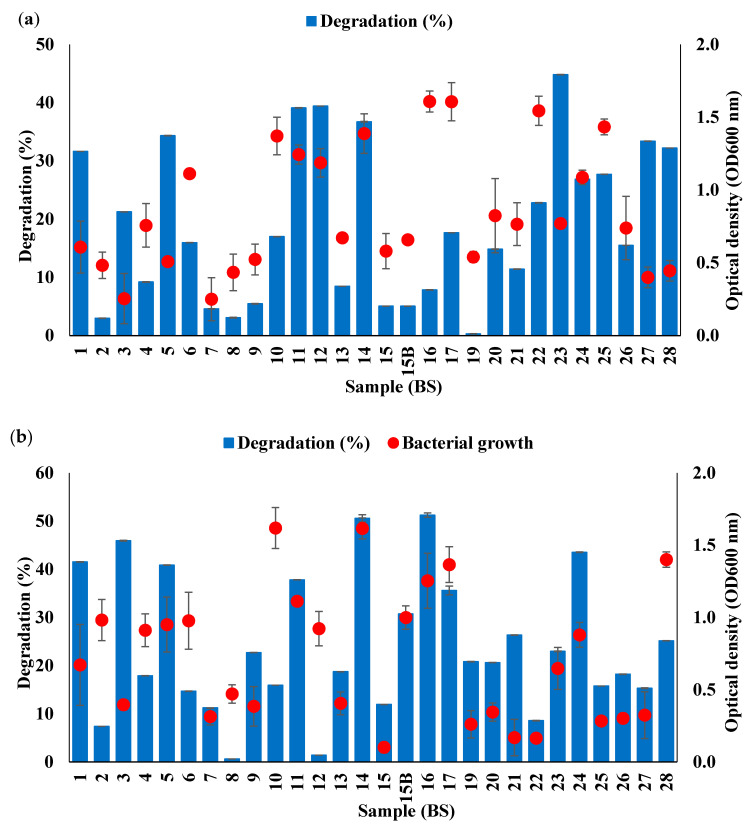
Percentage of (**a**) WCO and (**b**) PCO degradation by Antarctic bacterial consortia obtained from 28 soil samples over a 5 days incubation period.

**Figure 2 life-11-00456-f002:**
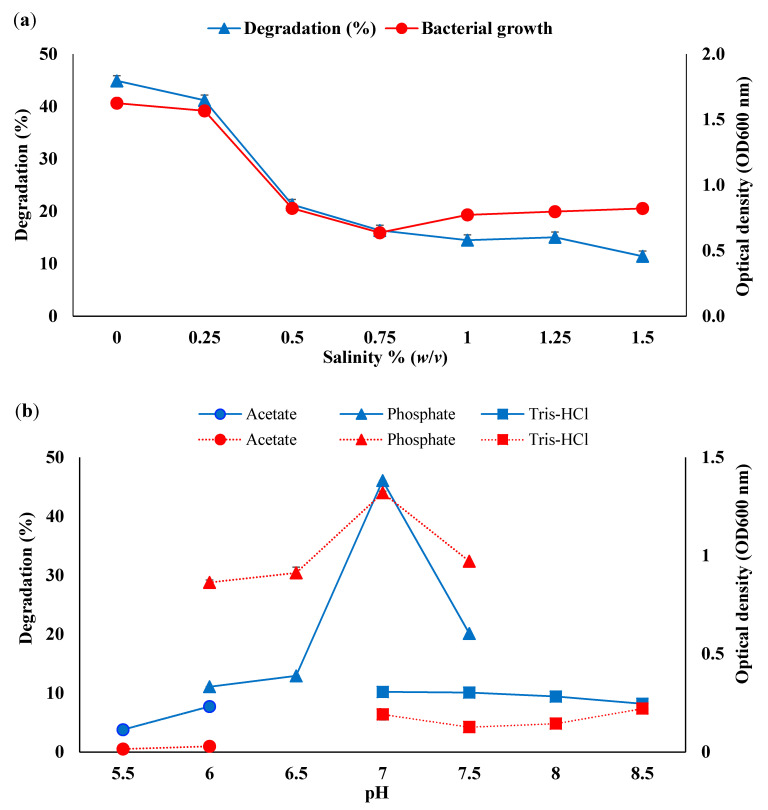
Effect of (**a**) salt concentration, (**b**) pH (dotted lines: acetate, solid lines: phosphate, dashed lines: Tris-HCl), (**c**) different nitrogen sources, (**d**) ammonium sulphate concentration, (**e**) temperature, (**f**) yeast extract, (**g**) initial substrate concentration on bacterial growth (filled circles) and WCO degradation (filled triangles).

**Figure 3 life-11-00456-f003:**
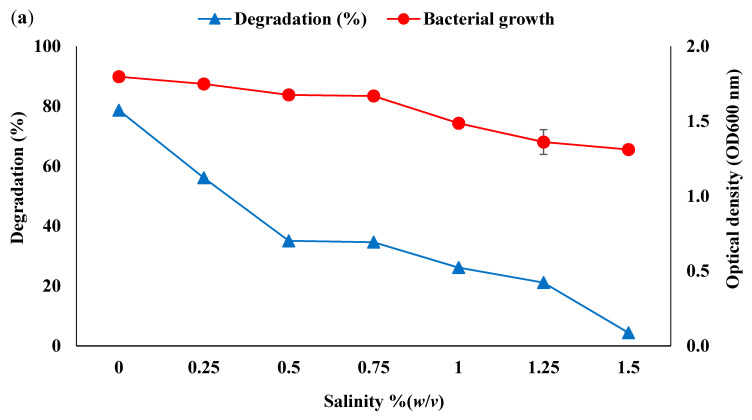
Effect of (**a**) salt concentration, (**b**) pH (dotted lines: acetate, solid lines: phosphate, dashed lines: Tris-HCl), (**c**) different nitrogen sources, (**d**) ammonium sulphate concentration, (**e**) temperature, (**f**) yeast extract, (**g**) initial substrate concentration on bacterial growth (filled circles) and PCO degradation (filled triangles/grey squares).

**Figure 4 life-11-00456-f004:**
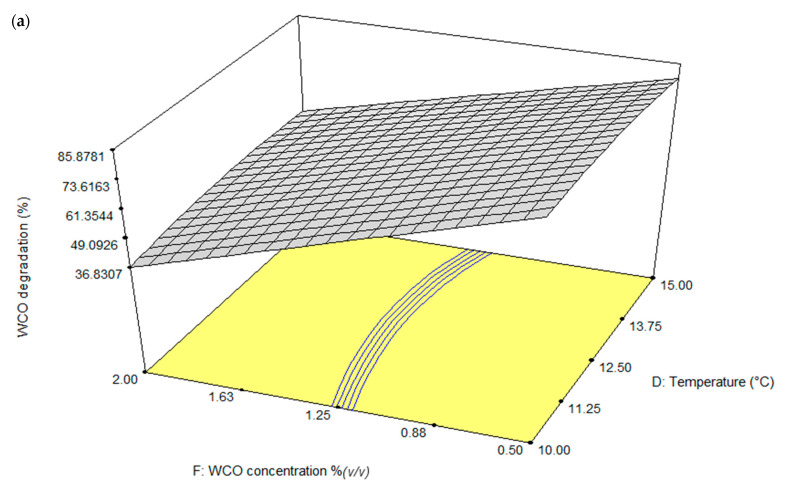
Three-dimensional (3D) response surface plots for the significant factors identified in CCD as influencing degradation of WCO ((**a**): temperature and initial WCO concentration) and PCO degradation ((**b**): pH and initial PCO concentration, (**c**): (NH_4_)_2_SO_4_ concentration and yeast extract, (**d**): (NH_4_)_2_SO_4_ concentration and initial PCO concentration) by the BS14 Antarctic bacterial consortium.

**Table 1 life-11-00456-t001:** Range and level of variables affecting WCO and PCO degradation by the selected Antarctic bacterial consortium (BS14) assessed using the Plackett-Burman design.

Factors	Name	Unit	Experimental Value		
			Low (−1)	Centre (0)	High (+1)
A	Salinity	% (*w*/*v*)	0.00	0.13	0.25
B	pH	-	7.00	7.25	7.50
C	(NH_4_)_2_SO_4_ concentration	g/L	0.50	0.75	1.00
D	Temperature	°C	10.0	12.5	15.0
E	Yeast extract	g/L	0.75	1.00	1.25
F	Initial oil concentration	% (*v*/*v*)	0.50	1.25	2.00

**Table 2 life-11-00456-t002:** Analysis of variance (ANOVA) for Plackett-Burman design (PBD) for WCO degradation.

Source	Sum of Squares	DF	F Value	Prob > F
Model	8404.79	6	2190.80	0.0005 ***
A	33.73	1	79.13	0.0124 *
B	17.16	1	40.26	0.0239 *
C	32.69	1	76.70	0.0128 *
D	3048.79	1	7152.30	0.0001 ***
E	63.29	1	63.29	0.0067 **
F	2349.43	1	2349.43	0.0002 ***
Residual	149.29	5		
Cor Total	8405.65	11		
*R*-squared	0.9999		Pred *R*-squared	N/A
Adj *R*-squared	0.9994		Adeq Precision	128.8396

A: salinity (% *w*/*v*), B: pH, C: (NH_4_)_2_SO_4_ concentration (g/L), D: temperature (°C), E: yeast extract (g/L), F: WCO concentration (% *v*/*v*) * *p* < 0.05, ** *p* < 0.01, *** *p* < 0.001.

**Table 3 life-11-00456-t003:** Analysis of variance (ANOVA) for Plackett-Burman design (PBD) for PCO degradation.

Source	Sum of Squares	DF	F value	Prob > F
Model	13,637.68	6	591.70	0.0001 ***
A	10.24	1	3.56	0.1558
B	85.28	1	29.60	0.0122 *
C	51.68	1	17.94	0.0241 *
D	26.10	1	9.06	0.0572
E	274.27	1	95.20	0.0023 **
F	4402.62	1	1528.15	<0.0001 ***
Residual	298.30	5		
Cor. Total	13,646.32	11		
*R*-squared	0.9994		Pred *R*-squared	0.9827
Adj *R*-squared	0.9977		Adeq Precision	55.992

A: salinity (% *w*/*v*), B: pH, C: (NH_4_)_2_SO_4_ concentration (g/L), D: temperature (°C), E: yeast extract (g/L), F: WCO concentration (% *v*/*v*) * *p* < 0.05, ** *p* < 0.01, *** *p* < 0.001.

**Table 4 life-11-00456-t004:** Analysis variance (ANOVA) for central composite data design (CCD) for WCO degradation response.

Source	Sum of Squares	DF	F value	Prob > F
Model	33,527.03	27	11.32	<0.0001 ***
A	4.45	1	0.041	0.8412
B	24.02	1	0.22	0.6418
C	321.12	1	2.93	0.0930
D	36.49	1	0.33	0.5666
E	459.53	1	6.15	0.0153 *
F	35,966.50	1	481.58	<0.0001 ***
A^2^	0.066	1	6.001 × 10^−4^	0.9805
B^2^	54.77	1	0.50	0.4829
C^2^	12.37	1	0.11	0.7383
D^2^	87.52	1	0.80	0.3758
E^2^	579.62	1	7.76	0.0067 **
F^2^	211.53	1	1.93	0.1708
AB	0.32	1	2.947 × 10^−3^	0.9569
AC	0.67	1	6.137 × 10^−3^	0.9379
AD	0.023	1	2.108 × 10^−4^	0.9885
AE	13.32	1	0.12	0.7289
AF	26.56	1	0.24	0.6247
BC	65.03	1	0.59	0.4448
BD	142.47	1	1.30	0.2596
BE	42.90	1	0.39	0.5344
BF	5.75	1	0.052	0.8197
CD	9.02	1	0.082	0.7754
CE	15.89	1	0.14	0.7050
CF	1.38	1	0.013	0.9111
DE	86.64	1	0.79	0.3782
DF	565.20	1	7.57	0.0074 **
EF	2.30	1	0.021	0.8853
Residual	4071.35	53		
Lack of Fit	4071.45	49	4.94	0.0642
Pure Error	0.00	4	0.00	
Cor Total	43,246.92	80		
*R*-squared	0.8688		Pred *R*-squared	0.8416
Adj *R*-squared	0.8618		Adeq Precision	55.8624

A: salinity (% *w*/*v*), B: pH, C: (NH_4_)_2_SO_4_ concentration (g/L), D: temperature (°C), E: yeast extract concentration (g/L), F: initial WCO concentration (% *v*/*v*) * *p* < 0.05, ** *p* < 0.01, *** *p* < 0.00.

**Table 5 life-11-00456-t005:** Analysis of variance (ANOVA) for central composite data design (CCD) for PCO degradation.

Source	Sum of Squares	DF	F value	Prob > F
Model	8831.46	14	8.28	0.0001 ***
A	555.71	1	7.16	0.0145 *
B	0.30	1	3.927 × 10^−3^	0.9509
C	335.79	1	4.41	0.0531
D	434.13	1	5.59	0.0283 *
A^2^	1622.05	1	20.89	0.0002 ***
B^2^	1466.52	1	18.89	0.0003 ***
C^2^	301.62	1	3.88	0.0627 *
D^2^	3105.00	1	39.99	<0.0001 ***
AB	0.018	1	2.413 × 10^−4^	0.9878
AC	56.01	1	0.74	0.4047
AD	1363.48	1	17.56	0.0005 ***
BC	603.67	1	7.77	0.0113 *
BD	477.31	1	6.15	0.0222 *
CD	17.89	1	0.23	0.6350
Residual	1142.88	15		
Lack of Fit	821.09	10	1.28	0.4161
Pure Error	321.79	5		
Cor Total	9974.34	29	8.28	
*R*-squared	0.8443		Pred *R*-squared	0.6001
Adj *R*-squared	0.7743		Adeq Precision	10.4597

A: pH, B: (NH_4_)_2_SO_4_ concentration (g/L), C: yeast extract concentration (g/L), D: initial PCO concentration (% *v*/*v*) * *p* < 0.05, ** *p* < 0.01, *** *p* < 0.001.

**Table 6 life-11-00456-t006:** Validation of predicted response surface model.

Type of Oil	Degradation (%)		*p* Value	Efficiency (%)
	Expected Value	Actual Value		
WCO	44.84	44.96	0.985	99.73
PCO	46.90	50.39	0.801	93.07

## Data Availability

Not applicable.

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
