# Peer review of "The Use of Response Surface Methodology as a Statistical Tool for the Optimisation of Waste and Pure Canola Oil Biodegradation by Antarctic Soil Bacteria"

_life, 2021, doi:10.3390/life11050456_

Round 1
Reviewer 1 Report
In this manuscript, the authors optimized various parameters for the antarctic soil bacteria-based degradation of waste and pure canola oil. Though the topic itself is interesting, there seem to be several issues the authors must address. A major revision is, therefore, recommended prior to further considerations.
- (Introduction) In the introduction part, the authors well discussed the limitations of previous studies. However, what this study is going to deal with seems to be lacking. It is therefore recommended to introduce more about the scope of this study.
- In line with the previous comment, please discuss how your optimized results could be potentially implemented in real field applications, especially if the optimized conditions are different from those of the target areas (antarctic area).
- (Materials and methods) Please more specifically provide the information of sampling spots (temperature, pH, etc.) since this information will be closely related to the optimal conditions.
- (Section 2.2) Why were the resting cells cultured for such a long time of 48 days?
- Though this study discussed their results with several references, it would be more convincing if the authors could provide their own supporting results. First, please compare the composition of waste canola oil and pure canola oil through GC analysis to reveal the reason for the differences in degradation efficiencies. In addition, for the consortium BS14, it is recommended to perform NGS analysis. Without knowing which species are involved in the efficient degradation, the findings of this study will be irreproducible.
Author Response
Comment 1
(Introduction) In the introduction part, the authors well discussed the limitations of previous studies. However, what this study is going to deal with seems to be lacking. It is therefore recommended to introduce more about the scope of this study.
Answer: The introduction has been explained in detail. Numerous sentences about the scope of this study were added in the introduction part. Page 2 and 3. Comment 2 In line with the previous comment, please discuss how your optimized results could be potentially implemented in real field applications, especially if the optimized conditions are different from those of the target areas (antarctic area). Answer: Some points on the application in Antarctic using this optimized result has been added in the introduction part on Line 109 to 121. Page 3. Comment 3 (Materials and methods) Please more specifically provide the information of sampling spots (temperature, pH, etc.) since this information will be closely related to the optimal conditions. Answer: We agreed that the information is important as we are studying the optimal conditions of the bacteria from that area. However, the information of the samples (BS1 until BS28) obtained were not completed. The samples were collected during January 2019. Only the information of the sample for BS14 (information collected only temperature and location was recorded during sampling) has been added in discussion part, Line 355 to 358 and 362 to 367. Page 9. Comment 4 (Section 2.2) Why were the resting cells cultured for such a long time of 48 days? Answer: There is typo in this section. The “48 d” was changed to “4 d” on Line 144. Page 3 Comment 5 Though this study discussed their results with several references, it would be more convincing if the authors could provide their own supporting results. First, please compare the composition of waste canola oil and pure canola oil through GC analysis to reveal the reason for the differences in degradation efficiencies. In addition, for the consortium BS14, it is recommended to perform NGS analysis. Without knowing which species are involved in the efficient degradation, the findings of this study will be irreproducible. Answer: We acknowledge the suggestion that you give on this part is valuable on this study. Unfortunately, the results on GC analysis could not be provided for current time. Our samples for this analysis still in que because of COVID-19 pandemic, it will take a long time to get the GC result. We only can cite past papers on the composition of oil and make a possibility on the oil for pure and waste canola oil. New points have been added on the Line 273 to 285. Page: 7-8.
Meanwhile, for identification of bacterial consortium BS14, I also could not give it right now since we are currently doing it and intend to submit to the other journal for identification using metagenomics analysis, and eventually could cited/referred this paper after completing the identification part later. We do understand the crucial of knowing the bacteria species involve in the biodegradation of oil, however We did mention or cited the past research on the possibilities of the bacteria involve in biodegradation oil in Antarctic itself (Line 105 to 107), and most of the bacteria involved in the biodegradation of oil are the same in the Antarctic according to the past studies. I also emphasized on the identified Antarctic bacteria in the results and discussions part as for new points to clarify this question (Line 252 to 268). Page: 3 & 6.
Reviewer 2 Report
Thank you for the opportunity to review the article entitled The use of Response Surface Methodology as a Statistical Tool for the Optimization of the Waste and Pure Canola Oil Biodegradation by Antarctic Soil Bacteria. The article is very interesting and quite well prepared. I only have two comments about it, which I have presented below. The article will be suitable for publication in the life journal after taking them into account.
- In my opinion introduction section should be extended because it is too general.
- How and from what depth were soil samples taken?
Author Response
Comment 1
In my opinion introduction section should be extended because it is too general.
Answer: The introduction has been explained in detail after correction. Numerous sentences were added. Page: 2-3
Comment 2
How and from what depth were soil samples taken?
Answer: We did explain in the section of 2.1 on how the soil samples were taken. The detail on the depth of soil collected has been added in this section on the line 127 to 128. Page: 3.
Round 2
Reviewer 1 Report
The quality of the manuscript seems to be improved.